# Lack of Association between Serum Chitotriosidase Activity and Arterial Stiffness in Type 2 Diabetes without Cardiovascular Complications

**DOI:** 10.3390/ijms242115809

**Published:** 2023-10-31

**Authors:** Luca D’Onofrio, Rocco Amendolara, Carmen Mignogna, Gaetano Leto, Lida Tartaglione, Sandro Mazzaferro, Ernesto Maddaloni, Raffaella Buzzetti

**Affiliations:** 1Department of Experimental Medicine, “Sapienza” University of Rome, 00161 Rome, Italy; luca.donofrio@uniroma1.it (L.D.); rocco.amendolara@uniroma1.it (R.A.); carmen.mignogna@uniroma1.it (C.M.); ernesto.maddaloni@uniroma1.it (E.M.); 2Diabetes Unit, Department of Medical-Surgical Sciences and Biotechnologies, Santa Maria Goretti Hospital, Sapienza University of Rome, 04100 Latina, Italy; gaetano.leto@uniroma1.it; 3Department of Translational and Precision Medicine, Sapienza University of Rome, 00185 Rome, Italy; lida.tartaglione@uniroma1.it (L.T.); sandro.mazzaferro@uniroma1.it (S.M.)

**Keywords:** type 2 diabetes, cardiovascular disease, arterial stiffness, pulse wave velocity, chitotriosidase

## Abstract

Chitotriosidase (CHIT), a mammalian chitinase secreted by neutrophils and activated macrophages, is increased in both cardiovascular disease (CVD) and type 2 diabetes (T2D). Arterial stiffness rises early in T2D and increases the risk of CVD. The aim of this study is to evaluate CHIT activity as an early biomarker of arterial stiffness in people with T2D free from overt vascular complications. In this cross-sectional study, arterial stiffness as measured using standard pulse wave velocity (PWV) was evaluated in 174 people with T2D without overt vascular disease. Then, we measured CHIT serum activity with an electrochemiluminescence assay in two subgroups of participants: 35 with the highest (high-PWV) and 40 with the lowest (low-PWV) PWV values. CHIT activity was no different between the low-PVW and high-PWV groups (12.7 [9.6–17.9] vs. 11.4 [8.8–15.0] nmol/mL/h, respectively). Compared with the low-PWV group, the high-PWV participants were older (*p* < 0.001); had a longer duration of diabetes (*p* = 0.03); higher ankle–brachial index ABI (*p* = 0.04), systolic blood pressure (*p* = 0.002), diastolic blood pressure (*p* = 0.005), fasting blood glucose (*p* = 0.008), and HbA1c (*p* = 0.005); and lower eGFR (*p* = 0.03) and body mass index (BMI) (*p* = 0.01). No association was present with sex, duration of diabetes, age, BMI, peripheral blood pressure, laboratory parameters, and glucose-lowering medications or ongoing antihypertensive therapy. Although no association was found, this study provides novel data about the association of CHIT activity with CVD, focusing on a specific outcome (arterial stiffness) in a well-defined population of subjects with T2D without established CVD.

## 1. Introduction

Type 2 diabetes mellitus (T2D) is a burgeoning worldwide health issue. Long disease duration, swinging glycemic levels, male sex, underlying comorbidities, and pre-existing modifiable risk factors negatively impact endothelial function [1,2]. All of these lead to the overt vascular complications of T2D [3,4], leading to a reduction in the quality of life and life expectancy of affected subjects. Arterial stiffness is an independent predictor of cardiovascular morbidity and mortality both in the general population and in individuals with T2D [5,6,7], exponentially increasing the risk of cardiovascular disease (CVD). Endothelial dysfunction is recognized as a key pathophysiological mechanism linking diabetes and arterial stiffness [8]. In this context, the chitotriosidase (CHIT) enzyme is regarded as a relevant player in immune response, as it is expressed by neutrophils and activated macrophages in response to a wide range of pro-inflammatory stimuli [9,10]. CHIT belongs to the human glycol-hydroxylase 18 family of enzymes, also known as chitinase, with the capacity to degrade chitin and chitin-containing pathogens [9,11]. While it represents a key function in the immune response, CHIT also takes part in several pathological conditions characterized by a marked inflammatory response, for example, osteoporosis [12], as we previously demonstrated, but especially atherosclerosis [13,14]. In this regard, serum CHIT activity is significantly increased in individuals with atherosclerosis, and it is related to the severity of the atherosclerotic lesion, suggesting a possible role as an atherosclerotic marker [14]. CHIT is also involved in the pathological process of cardiovascular complications in people with T2D, as suggested by increased levels of CHIT in subjects with T2D and overt CVD [15,16]. Furthermore, it is also known that CHIT activity is related to circulating markers of endothelial dysfunction in people with T2D without CVD [17]. Increased vascular stiffness, as measured using standard arterial pulse wave velocity (PWV), seems to occur very early, even in the prediabetic stages [18,19], pointing to PWV as an early biomarker of vascular disease. Based on this evidence, we hypothesized that CHIT may be regarded as a possible marker of early vascular impairments in settings of uncomplicated T2D. Therefore, the aim of this study was to investigate the CHIT activity in people with T2D free from overt vascular complications with a high or low grade of arterial stiffness measured using standard PWV.

## 2. Results

### 2.1. Population Features

After laboratory measurements, 10 individuals were excluded from further analysis as they showed a serum activity of CHIT < 5 nmol/mL/h, and one patient was excluded as an outlier (Appendix A). A total of 75 Caucasian individuals affected by T2D (64% males aged 55 ± 5.4 years) were included in this study. The duration of diabetes was 5 [3–8] years and HbA1c was 6.2 [5.8–6.7]% (44 [40–50] mmol/mol). Regarding ongoing glucose-lowering therapy, 87% of participants were on diet plus oral hypoglycemic agents (73% were on metformin alone, 11% on metformin plus others glucose-lowering drugs, 3% on dipeptidyl peptidase-4 alone, and 1% on pioglitazone alone), while the remaining were on diet alone (12%). A history of arterial hypertension was present in 67%, while 45% had dyslipidemia and 45% had obesity. Table 1 summarizes the study population characteristics stratified by the average of the haPWV values.

Briefly, participants within the high-PWV group were older (*p* < 0.001); had a longer duration of diabetes (*p* = 0.03); higher SBP (*p* = 0.002), DBP (*p* = 0.005), fasting blood glucose (*p* = 0.008), and HbA1c (*p* = 0.005); and lower BMI (*p* = 0.013) and eGFR (*p* = 0.03) compared to subjects with lower values of PWV. Participants in the high-PWV group were more likely to be affected by hypertension than those in the low-PWV group (*p* = 0.005). The two groups also differed in ABI values, and, in particular, subjects with a high PWV showed an increased ABI (*p* = 0.04), supporting the increased stiffness in this group of subjects. No difference was found between groups in terms of sex, smoking habits, dyslipidemia, obesity, glucose-lowering medications, and ongoing antihypertensive therapy.

### 2.2. Chitotriosidase Activity and Parameters of Arterial Stiffness

In the whole population, the serum level of CHIT activity was 12.4 [9.5–15.6] nmol/mL/h. No significant difference in CHIT activity was found between the low- (12.7 [9.6–17.9] nmol/mL/h) and high-PWV (11.4 [8.8–15] nmol/mL/h) groups (Figure 1). Regarding comorbidities, no difference was found in CHIT activity stratifying the population by the presence of hypertension, dyslipidemia, or obesity (Appendix A). Neither tobacco use, physical activity, nor hypoglycemic treatment influenced CHIT activity.

In the whole population, no association was present between CHIT activity and sex, duration of diabetes, age, BMI, peripheral blood pressure, physical activity, laboratory parameters, glucose-lowering medications, or ongoing antihypertensive therapy.

## 3. Discussion

The present study aimed to provide insights into the potential involvement of CHIT as an early biomarker of arterial stiffness, taking advantage of a well-characterized cohort of individuals with T2D without overt cardiovascular complications. Our results indicated that CHIT activity is not associated with parameters of arterial stiffness, such as haPWV.

Despite the negative results, this study contributes to a debated field of research regarding the clinical and pathological values of novel markers of early cardiovascular damage in T2D [20,21,22]. In this regard, few previous studies have investigated the role of circulating human CHIT in the early phases of diabetes and arterial damage. In particular, Sonmez and colleagues reported that plasma CHIT activity was significantly higher in people with newly diagnosed, uncomplicated, and untreated T2D than in those without diabetes [17]. In particular, the authors found that, after adjustments for covariates, including age, BMI, blood pressure, and lipid profile, CHIT activity was an independent determinant of plasma asymmetric dimethylarginine (ADMA) levels, a surrogate marker of arterial stiffness [17]. Furthermore, in a prospective observational study conducted on 119 subjects with T2D, Naka et al. demonstrated that higher levels of Chitinase 3-like protein 1 (YKL-40), another human chitinase, were associated with increased aortic PWV in individuals with T2D without established CVD [23]. In addition, several data have been published regarding the positive association between the serum levels of YKL-40 and carotid–femoral PWV in diseases other than diabetes, such as ankylosing spondylitis, essential hypertension, and rheumatoid arthritis [24,25,26].

However, the majority of studies published so far have highlighted an elevation in CHIT activity in T2D subjects with cardiovascular disease when compared with those without vascular complications [15,16]. Under these conditions, an accumulation of lipid-laden macrophages is likely to have occurred during the gradual progression of atherosclerosis, leading to an increase in CHIT enzyme activity. Nonetheless, the same studies did not confirm a significant difference between individuals with T2D free of vascular complications and healthy volunteers [15,16], in line with our results.

In contrast, our findings did not support an association between the serum CHIT activity and PWV or other parameters of early vascular damage even in subjects with the highest arterial stiffness. However, to date, this is the first effort to evaluate the possible involvement of CHIT enzyme as an early biomarker involved in the pathogenesis of arterial stiffness in a well-characterized Caucasian population having T2D without cardiovascular complications. The contrasting results could be explained by the differences in the glyco-metabolic control [17] and age [27]. To date, there are no data correlating the enzymatic activity of YKL-40 to the parameters of arterial stiffness. Furthermore, without longitudinal studies, the available evidence does not allow the exclusion of a time-dependent variation in the two chitinases (CHIT and YKL-40) during the various stages of vascular damage. It may be suggested that CHIT is involved in the early stages of disruption of the endothelium homeostasis, depending on poor glycemic control [15,17], leading to an increase in arterial stiffness, which in turn results in an increase in YKL-40 levels. At the later stages of atherosclerotic plaque development, increased inflammation and the activation of an increasing number of macrophages may result in an enhancement in both chitinases. However, no longitudinal study has demonstrated causality between the chitinases and the early disruption of vascular health.

Our findings should be interpreted within the context of the strengths and limitations of this study. To the best of our knowledge, the present study is the first effort to evaluate the possible role of the serum activity of the CHIT enzyme in the arterial stiffness of vessel walls in a population of subjects with T2D without CVD. Our study included participants who underwent a comprehensive measurement of the parameters of arterial rigidity and clinical assessment. Furthermore, we enrolled participants who did not show any sign of cardiovascular complications in order to properly estimate the involvement of serum CHIT activity in the early phases of vascular derangements. However, some limitations must be acknowledged. First, the cross-sectional design did not allow us to make any inference regarding causality. Second, inherited defects of the CHIT gene might have confounded our results. The presence of a null allele in the CHIT gene has a prevalence of 3–4% in the general population [28]. However, to avoid this contingency, individuals with a serum CHIT activity below 5 nmol/mL/h were excluded from the present study. Further large-scale evaluations should include different degrees of cardiometabolic risk profiles to properly assess the role of CHIT activity over the different stages of the vascular complications. Fourth, the reproducibility of the haPWV measurement was not tested, but the procedure was well standardized, and the same technician carried out all the haPWV assessments with controlled conditions. Thus, this limitation did not affect our results. We selected two groups with extreme values of haPWV in order to maximize the probability of finding differences in the levels of CHIT. Then, a small cohort of subjects with T2D was enrolled in the final analysis, but, following the sample size evaluation, we can say that this study was adequately powered to test its primary aim. Finally, the restricted age range of our study cohort probably prevented us from finding an increase in age-dependent CHIT enzymatic activity, as previously reported [27].

In conclusion, our findings demonstrated that serum CHIT activity was not associated with arterial stiffness in a well-characterized population of individuals with T2D without overt cardiovascular complications. Nonetheless, future efforts should be made to investigate the importance of chitinases in cardiovascular risk stratification.

## 4. Materials and Methods

### 4.1. Research Design and Population

This is a cross-sectional pilot study, in which we compare serum CHIT levels in subjects with high (high-PWV) and low PWV (low-PWV). We enrolled participants from May 2017 to December 2019 referred to the outpatient clinics of the Diabetology Unit of Santa Maria Goretti Hospital, Polo Pontino Sapienza University of Rome (Italy), and the outpatient clinic of Diabetology Unit of Policlinico Umberto I, Rome (Italy). We consecutively screened 174 subjects with T2D who underwent a comprehensive evaluation of arterial stiffness status. Specifically, we measured the haPWV of all screened subjects (*n* = 174) and enrolled participants with the highest (high-PWV) and lowest (low-PWV) values of PWV. The cutoff used for assigning the study participants to the high or low group was estimated based on the median haPWV value (7.9 [7.3–8.5] m/s). In particular, according to the sample size evaluation, 43 subjects per group were consecutively enrolled, for a total of 86 participants. A detailed description of the enrolment strategy is summarized in the Appendix A. The participants were carefully screened and enrolled according to the following inclusion criteria: (1) confirmed diagnosis of T2D according to the American Diabetes Association guidelines [29]; (2) males and females aged 18 to 65 years; (3) ability and willingness to provide written and informed consent. Participants were excluded from this study if they had a previous history of coronary heart disease, retinopathy, neuropathy, or other vascular disease; an estimated glomerular filtration rate (eGFR) <60 mL/min × 1.73 m^2^; ongoing treatment with exogenous insulin or sulfonylureas; traumatic bone fractures of recent onset (<6 months); diagnosis of autoimmune diabetes and other autoimmune diseases; diagnosis of Gaucher’s disease; resistant arterial hypertension (blood pressure above the target range set by current guidelines despite concurrent use of >3 antihypertensive drugs of different classes); recent history of neoplasia (<5 years) or kidney transplant; or serum CHIT activity lower than 5 nmol/mL/hour, in order to avoid the presence of participants carrying genetic variants associated with severely reduced or totally absent enzymatic activity.

### 4.2. Clinical Assessment

The following anthropometric data were recorded from the clinical charts: sex, age, duration of diabetes, weight, and height. A current smoker was defined as smoking at least one cigarette/day in the last year. The body mass index (BMI) was calculated as weight in kilograms divided by height in square meters (kg/m^2^). The obesity state was inferred as a BMI ≥ 30 kg/m^2^. Blood pressure was measured on the day of clinical examination. The participants were kept resting in a sitting position for at least 5–10 min in a fasted condition, avoiding tobacco use or the intake of coffee or any stimulating beverage before the evaluation. The average of three consecutive measurements was used for the analysis. Hypertension was defined as a systolic blood pressure ≥140 mmHg or a diastolic blood pressure ≥90 mmHg or any use of antihypertensive drugs. Dyslipidemia was considered as a total cholesterol level of >220 mg/dL and/or a triglyceride level of >150 mg/dL or ongoing treatment with lipid-lowering medication. The past medical history was retrieved from medical charts.

### 4.3. Arterial Stiffness Measurement

The parameters of arterial stiffness (e.g., haPWV and ABI) were measured noninvasively using the VaSera 1500 system (Fukuda Denshi) with a standard procedure [30]. Briefly, while subjects were resting in the supine position, four blood pressure cuffs were wrapped on the four extremities. Electrocardiography electrodes were attached to the upper arms, and a microphone was placed on the sternum in the second intercostal space. After the patient was stabilized in the supine position for 5 min, electrocardiography and phonocardiography were monitored, and the blood pressure and the waveforms of the brachial and ankle arteries were measured. The ABI was obtained by dividing the ankle systolic blood pressure (SBP) by the arm SBP. The heart–ankle PWV (haPWV) was obtained using the following formula: haPWV = L/T, where L is the vascular length between the aortic valve and the ankle indirectly calculated from the individual height of the patient (0.77685 × height (cm) − 1.7536) and T is the time taken for the pulse wave to propagate from the aortic valve to the ankle. The average of the right and left haPWV was used for the analysis. The result was expressed in meter/seconds (m/s). According to the average of the PWV values measured in our study cohort, we divided the population into the lowest (low-PWV, *n* = 43) and highest (high-PWV, *n* = 43) PWV subgroups.

### 4.4. Laboratory Evaluations

After an overnight fast (8–10 h), the fasting blood glucose (FBG), glycated hemoglobin (HbA1c), total cholesterol, triglycerides, high-density lipoprotein cholesterol (HDL-c), creatinine, high-sensitivity C-reactive protein (hsCRP), uric acid, calcium, phosphate, vitamin D, albumin, and parathyroid hormone (PTH) were measured with standard laboratory procedures. The low-density lipoprotein cholesterol (LDL-c) concentrations were estimated with the use of Friedewald et al.’s formula [31]. The eGFR was estimated according to Levey et al. [32].

At the time of the blood draw, an additional sample was gathered, subsequently stored at −80 °C and preserved for CHIT activity evaluation. The CHIT activity was measured by incubating 5 uL of serum with 100 uL of 0.022 mM 4-methylumbelliferyl-fl-D-N,N,N’-triacetylchitotriose (Sigma Chemical Co., St. Louis, MO) as a substrate in a citrate/phosphate buffer (0.1/0.2 M), pH 5.2, at 37 °C as described by Hollak et al. [33]. After 15 min, the reaction was stopped with 2 mL of 0.5mol/L Na2CO3 NaHCO3 buffer, pH 10.7. Fluorescent 4-methylumbelliferone was measured with the F-2500 Fluorescence Spectrophotometer (Hitachi Corp., Chiyoda, Japan) at 450 nm. The CHIT activity is indicated as nanomoles of substrate hydrolyzed per milliliter per hour (nmol/mL/h).

### 4.5. Statistical Analysis

The continuous variables are presented as means ± SD or median [25th–75th percentiles], and the discrete variables are expressed as frequencies and percentages. The Shapiro–Wilk test was used to evaluate the parametric distribution of continuous variables. Values greater than the 3-fold interquartile range (IQR) were considered as extreme outliers and excluded from the analysis. Student’s *t*-test or Kruskal–Wallis’ test was used to evaluate differences in continuous variables between groups, as appropriate. The chi-squared test was used to test differences in the distribution of categorical variables between the study groups. The correlation between the serum activity of CHIT and the clinical/biochemical data was tested by Pearson’s or Spearman’s test, based on the distribution. Two-sided tests at the 0.05 level of significance were used for all statistical comparisons. Statistical analysis was performed with IBM SPSS software (version 27.0.1), and Prism 8 software was used for graphical representations.

#### Sample Size Calculation

To properly assess the relation between the CHIT activity and the parameters of arterial rigidity, the sample size was calculated using Stata/SE 13.0 software. Based on a previous publication [17,34], the mean and SD of CHIT activity in people with T2D and healthy subjects were 88.1 ± 54.90 nmol/mL/h and 50.7 ± 37.11 nmol/mL/h, respectively. Setting α = 0.05 and β = 0.10 (90% power), the sample size needed to identify a significant difference of about 35 nmol/mL/h of CHIT between the two study groups is *n* = 35 patients per group. Furthermore, to avoid possible bias due to the dosage of CHIT, we decided to enroll at least 20% more subjects.

## Figures and Tables

**Figure 1 ijms-24-15809-f001:**
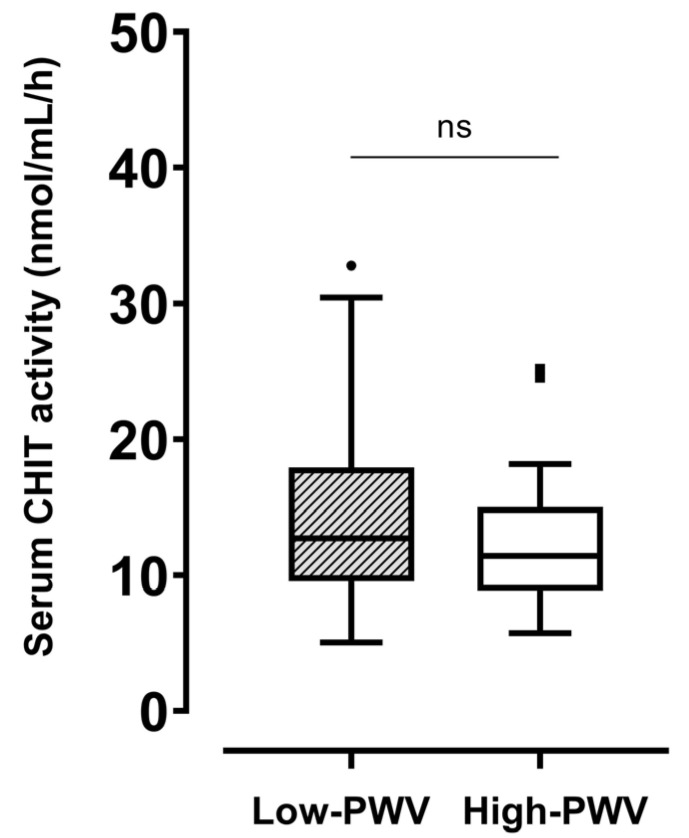
Differences in serum CHIT activity between participants with haPWV ≥ 8 m/s and those with haPWV < 8 m/s. CHIT, chitotriosidase; haPWV, heart–ankle pulse wave velocity. ns: not significant.

**Table 1 ijms-24-15809-t001:** Features of the study population stratified by the average of PWV values.

	**Low PWV** (***n* = 40**)	**High PWV**(***n* = 35**)	** *p* ** **-Value**
Age, y	52.3 ± 4.4	57.9 ± 5	**<0.001**
Males, *n* (%)	24 (60)	24 (69)	0.440
Duration of diabetes, y	5 [2–7]	6 [3–9]	**0.029**
BMI, kg/m^2^	30 [27.6–35.4]	27.1 [26–31.1]	**0.013**
SBP, mmHg	140 [129–153]	148 [140–165]	**0.002**
DBP, mmHg	86.1 ± 8.5	91.9 ± 9	**0.005**
haPWV, m/sec	7 ± 0.6	9.1 ± 0.6	**<0.001**
ABI	1.06 ± 0.1	1.1 ± 0.06	**0.04**
HbA1c, %	6.1 [5.6–6.5]	6.6 [6–7]	**0.005**
FBG, mg/dL	113 [102–129]	130 [113–139]	**0.008**
Total cholesterol, mg/dL	175.2 ± 31.9	169 ± 34.9	0.424
HDL-c, mg/dL	50.1 ± 11.7	51.1 ± 14.1	0.729
LDL-c, mg/dL	100.6 ± 25.8	93.4 ± 25.8	0.236
Triglycerides, mg/dL	115 [82–156]	104 [71–127]	0.256
Ca, mg/dL	9.6 ± 0.4	9.5 ± 0.5	0.514
P, mg/dL	3.5 ± 0.6	3.3 ± 0.5	0.229
Vitamin D, ng/mL	26 [16–34]	23 [13–30]	0.087
PTH, pg/mL	68 [59–97]	70 [55–95]	0.776
CRP, mg/L	0.13 [0.04–0.4]	0.15 [0.07–0.4]	0.555
Albumin (g/dL)	4.3 ± 0.3	4.4 ± 0.2	0.126
Creatinine (mg/dL)	0.8 [0.7–0.9]	0.8 [0.7–1]	0.483
eGFR (mL/min/1.73 m^2^)	98.3 [93.9–101]	94 [84–100]	**0.030**
Uric acid (mg/dL)	5.6 ± 1.4	5.4 ± 1.2	0.581
Chit activity (nmol/mL/h)	12.7 [9.6–17.9]	11.4 [8.8–15]	0.230
**Smoking status, *n*** (**%**)			
Current smoker	10 (25)	9 (26)	0.995
Former smoker	10 (26)	9 (28)	
Never	20 (50)	17 (46)	
**Past medical history, *n*** (**%**)			
History of hypertension	21 (51)	29 (83)	**0.005**
Dyslipidemia	17 (43)	17 (49)	0.598
Obesity	21 (51)	13 (37)	0.246
**Ongoing glucose-lowering therapy, *n*** (**%**)			
Metformin	26 (65)	27 (77)	0.313
Metformin + others	6 (15)	4 (11)	0.745
Dipeptidyl peptidase-4	0 (0)	2 (6)	0.209
Pioglitazone	0 (0)	1 (3)	0.461

Continuous variables are reported as means ± SD, or median (25th–75th percentiles) for variables with skewed distribution. Categorical variables are presented as frequencies and percentages. *P*-values are calculated using Student *t*-test, Mann–Whitney *U* test, chi-squared test, or Fisher’s exact test, as appropriate. Abbreviations: ABI, ankle–brachial index; haPWV, heart–ankle pulse wave velocity; BMI, body mass index; Ca, calcium; DBP, diastolic blood pressure; eGFR, estimated glomerular filtration rate; FBG, fasting blood glucose; HbA1c, glycosylated hemoglobin; HDL-c, high-density lipoprotein cholesterol; CRP, C-reactive protein; LDL-c, low-density lipoprotein cholesterol; P, phosphate; PTH, parathyroid hormone; SBP, systolic blood pressure.

## Data Availability

Data are available upon reasonable request to the corresponding author.

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
