# Peer review of "Lack of Association between Serum Chitotriosidase Activity and Arterial Stiffness in Type 2 Diabetes without Cardiovascular Complications"

_ijms, 2023, doi:10.3390/ijms242115809_

Round 1
Reviewer 1 Report
Comments and Suggestions for Authors
Congratulations on your work. I read your manuscript with great interest. However I have few comments
1. The structure of the manuscript should be corrected. You present results and discussion before study design and methods. Because of that it is hard to understand the study properly
2. How did you divided your study population into two groups based on PWV levels? It should be described properly how you determine cut off value.
3. The study group is not large, it should be highlighted in the study limitations.
Reviewer 2 Report
Comments and Suggestions for Authors
The manuscript titled “Lack of association of serum chitotriosidase activity with arterial stiffness in type 2 diabetes without cardiovascular complications” discusses the relationship between T2D and arterial stiffness, as well as the role of CHIT in immune response and its involvement in several pathological conditions characterized by a marked inflammatory response, including atherosclerosis. The title describes the contents of the study adequately. The statements also suggest that CHIT may be regarded as a possible marker of early vascular impairments in settings of uncomplicated T2D.
The study included participants who underwent a comprehensive measurement of the parameters of arterial rigidity and clinical assessment. Furthermore, participants who did not show any sign of cardiovascular complications were enrolled to properly estimate the involvement of serum CHIT activity in the early phases of vascular derangements.
The results indicated that CHIT activity is not associated with the parameters of arterial stiffness, such as haPWV. Despite the negative results, this study contributes to a debated field of research regarding the clinical and pathological values of novel markers of early cardiovascular damage in T2D. The abstract provides sufficient information about the study design, methods, and results. However, it does not justify the significance of the results.
Previous studies have investigated the role of circulating human CHIT in the early phases of diabetes and arterial damage. Turan et al. reported that plasma CHIT activity was significantly higher in people with newly diagnosed, uncomplicated and untreated T2D than those without diabetes (S. Sonmez et al., “Plasma chitotriosidase activity in metabolic syndrome,” Journal of Clinical Endocrinology & Metabolism, vol. 95, no. 8, pp. 3913-3918, 2010. https://doi.org/10.1210/jc.2009-2773).
Naka et al. demonstrated that higher levels of Chitinase3-like protein 1 (YKL-40), another human chitinase, were associated with increased aortic PWV in individuals with T2D without established CVD (Naka et al., “Chitinase 3-like protein predicts coronary events and revascularization in patients with suspected coronary artery disease: The coronary artery biomarker study (CABG-CTO),” Atherosclerosis, vol. 219, no. 1, pp. 221-226, 2011. https://doi.org/10.1016/j.atherosclerosis.2011.06.035).
Several data have been published regarding the positive association between the serum levels of YKL-40 and carotid-femoral PWV in diseases other than diabetes, such as ankylosing spondylitis, essential hypertension and rheumatoid arthritis.
The cross-sectional design did not allow the researchers to make any inference regarding causality.
Inherited defects of the CHIT gene might have confounded the results. The presence of a null allele in the CHIT gene has a prevalence of 3-4% in the general population. However, to avoid this contingency, individuals with serum CHIT activity below 5 nmol/ml/h were excluded from the present study.
Further large-scale evaluations should include different degrees of cardiometabolic risk profiles to properly assess the role of CHIT activity over the different stages of vascular complications.
The reproducibility of haPWV measurement was not tested, but the procedure seems to be standardized.
I regret to say that the content does not justify the urgency or significance of being presented as communication.
Round 2
Reviewer 2 Report
Comments and Suggestions for Authors
The manuscript has the potential to be strengthened by the addition of further experiments. Nonetheless, the authors have presented a compelling case for the significance of their negative results, indicating the importance of publishing to prevent publication bias. I would suggest accepting the manuscript in the present form. Best
Comments on the Quality of English LanguageNo major changes are required.